# Imeglimin Inhibits Macrophage Foam Cell Formation and Atherosclerosis in Streptozotocin-Induced Diabetic ApoE-Deficient Mice

**DOI:** 10.3390/cells14070472

**Published:** 2025-03-21

**Authors:** Ji Yeon Lee, Yup Kang, Ja Young Jeon, Seung Jin Han

**Affiliations:** 1Department of Endocrinology and Metabolism, Ajou University School of Medicine, Suwon 16499, Republic of Korea; jlee@aumc.ac.kr (J.Y.L.); twinstwins@ajou.ac.kr (J.Y.J.); 2Department of Physiology, Ajou University School of Medicine, Suwon 16499, Republic of Korea; kangy@ajou.ac.kr

**Keywords:** imeglimin, atherosclerosis, type 2 diabetes, macrophage, foam cell formation, cholesterol efflux

## Abstract

Atherosclerotic cardiovascular disease is a major complication of diabetes, whose progression is significantly accelerated by hyperglycemia. Imeglimin, a novel oral antidiabetic agent, has demonstrated efficacy in glucose control; however, its role in diabetes-related cardiovascular complications has not yet been fully explored. This study aimed to investigate the effects of imeglimin on foam cell formation and atherosclerosis in the context of diabetes. THP-1 macrophages were treated with oxidized low-density lipoprotein (LDL) and high glucose to induce foam cell formation in vitro. Additionally, ApoE^−/−^ mice with streptozotocin-induced diabetes were used to determine the effects of imeglimin in vivo by analyzing metabolic parameters and atherosclerotic plaque formation. Imeglimin inhibited macrophage-derived foam cell formation by promoting the expression of ATP-binding cassette transporters (ABC) A1 and ABCG1 and downregulating the expression of CD36. The effects of imeglimin on ABCG1 and CD36 expression regulation was mediated by AMPK. In diabetic ApoE^−/−^ mice, imeglimin reduced the atherosclerotic plaque area, decreased fasting glucose and LDL cholesterol levels, and upregulated ABCG1 expression in the liver and aorta. These findings suggest that imeglimin may have a preventive effect on foam cell formation and a therapeutic role in atherosclerosis progression in diabetic conditions.

## 1. Introduction

Atherosclerotic cardiovascular disease is a serious complication of diabetes and remains a leading cause of mortality and morbidity in affected individuals, significantly impacting their overall health and quality of life [1,2]. In diabetes, metabolic disturbances such as chronic hyperglycemia, dyslipidemia, oxidative stress, and chronic inflammation contribute to endothelial dysfunction, platelet activation, and vascular inflammation, ultimately accelerating the progression of atherosclerosis [3].

After infiltrating the endothelial barrier, macrophages accumulate in the arterial intima media, where they act as the primary source of foam cells in response to the proinflammatory activation of endothelial cells [4]. These foam cells exacerbate inflammation within the arterial wall, contributing to various pathological outcomes, including rupture, hemorrhage, and calcification [5]. Macrophages accumulate lipids and transform into foam cells upon uptake of oxidized low-density lipoproteins (ox-LDLs) via scavenger receptors such as the cluster of differentiation (CD) 36 [6]. Conversely, cholesterol efflux from the macrophages is critical for preserving cholesterol homeostasis. This efflux is mediated by the ATP-binding cassette transporters A1 (ABCA1) and G1 (ABCG1), which facilitate the transfer of cholesterol from foam cells to the extracellular acceptors apolipoprotein A-I (ApoA-I) and high-density lipoprotein (HDL), respectively [7]. In atherosclerosis, regulation of macrophage cholesterol handling is disrupted [4]. Therefore, strategies focused on minimizing foam cell formation and enhancing cholesterol efflux from macrophages are essential for preventing atherosclerosis.

Imeglimin, a novel oral glucose-lowering agent, was approved for the treatment of type 2 diabetes in Japan in 2021 [8]. It is the first drug in the glimins class and is marketed under the brand name TWYMEEG^®^. The approval was based on the results of several clinical trials, including pivotal Phase III studies, which demonstrated its efficacy and safety in managing blood glucose levels both as a monotherapy and in combination with other antidiabetic drugs [9,10,11]. Imeglimin improves glycemic control by increasing insulin secretion from pancreatic β-cells and enhancing insulin sensitivity in the liver and muscles, thereby offering a comprehensive approach to managing type 2 diabetes [12]. Since 2008, the US Food and Drug Administration has recommended that new drugs for type 2 diabetes undergo clinical trials to demonstrate cardiovascular safety, in addition to their glycemic benefits [13]. However, to date, no clinical studies have been conducted on the cardiovascular safety of imeglimin; thus, its cardiovascular safety profile remains unknown.

Several experimental studies have suggested that imeglimin may be beneficial for reducing the risk of atherosclerotic cardiovascular diseases. Imeglimin has been shown to protect human endothelial cells from oxidative stress-induced apoptosis and improve left ventricular dysfunction in a rat model of metabolic syndrome [14,15]. Additionally, recent studies have reported that imeglimin exerts anti-inflammatory effects by inhibiting NLRP3 inflammasome activation in macrophages and microglia cells [16,17]. However, the effects of imeglimin on foam cell formation from macrophages remain unclear. Consequently, we aimed to elucidate the effects of imeglimin on foam cell formation and related molecules in THP-1 macrophages under ox-LDL and high glucose (HG). Furthermore, we investigated whether imeglimin could reduce atherosclerotic plaque formation in diabetic ApoE^−/−^ mice.

## 2. Materials and Methods

### 2.1. Cell Culture and Treatment

The THP-1 human monocytic cell line (American Type Culture Collection, Manassas, VA, USA) was maintained in RPMI 1640 medium (Corning, Manassas, VA, USA) supplemented with 10% fetal bovine serum (FBS; Corning, Woodland, CA, USA) and 1% penicillin/streptomycin. Cells were incubated at 37 °C in a humidified atmosphere containing 5% CO_2_. For differentiation into macrophages, THP-1 cells were stimulated with 100 nM phorbol 12-myristate 13-acetate (PMA; Sigma–Aldrich, Saint Louis, MO, USA) for 48 h in 6-well culture plates. To establish an in vitro model of diabetic atherosclerosis, macrophage-differentiated THP-1 cells were exposed to ox-LDL (100 μg/mL, Invitrogen, Eugene, OR, USA) combined with HG (30 mM) for 24 h [18,19]. Prior to this treatment, cells were incubated with or without imeglimin (AdooQ Bioscience, Irvine, CA, USA) for 24 h. Where necessary, an AMPK inhibitor, compound C (10 μM, Cell Signaling Technology Inc., Boston, MA, USA), was applied for 24 h.

### 2.2. Cholesterol Efflux Assay

Cholesterol efflux was evaluated in macrophage-differentiated THP-1 cells using a commercial Cholesterol Efflux Assay Kit (Abcam, Cambridge, MA, USA), following the manufacturer’s protocol. Briefly, THP-1 cells in a 96-well plate were washed with serum-free medium and incubated with the labeling reagent for 90 min in a humidified incubator with 5% CO_2_, protected from light. After labeling, the medium was removed, the cells were treated with equilibration medium containing the appropriate concentrations of Reagents A and B, and incubated at 37 °C for 18 h, protected from light. Following incubation, the medium was removed, and the cells were treated with HDL cholesterol acceptors and incubated for 6 h. The culture medium was then collected, and cell monolayers were lysed at room temperature (20–25 °C) for 30 min.

Fluorescence intensities of the collected medium and lysates were recorded at excitation/emission wavelengths of 485 nm and 523 nm, respectively. The cholesterol efflux percentage was calculated as the fluorescence intensity of the medium relative to the total fluorescence in both the medium and the cell lysate.

### 2.3. Western Blotting

Cell and tissue samples were lysed in ice-cold CelLytic M lysis buffer (Sigma–Aldrich) supplemented with protease and phosphatase inhibitors (Cell Signaling Technology Inc.), followed by incubation on ice for 30 min. Lysates were clarified by centrifugation at 14,000× *g* for 30 min at 4 °C. Protein concentrations were determined using a bicinchoninic acid (BCA) assay kit (Thermo Fisher Scientific, Rockford, IL, USA). Equal amounts of protein (20 µg) were loaded onto NuPAGE™ 4–12% Bis-Tris gradient gels (NOVEX by Life Technologies, Winston-Salem, NC, USA) and electrophoresed before being transferred onto nitrocellulose membranes. Membranes were blocked using 5% bovine serum albumin at room temperature for 1 h, followed by overnight incubation at 4 °C with primary antibodies targeting CD36 (PA1-16813, 1:1000, Thermo Fisher Scientific), ABCA1 (NB400-105, 1:1000, Novus Biologicals, Littleton, CO, USA), ABCG1 (NB400-132, 1:1000, Novus Biologicals), AMPK (2532, 1:1000, Cell Signaling Technology), phosphorylated AMPK (pAMPK, 2535, 1:1000, Cell Signaling Technology), and β-actin (ab8227, 1:10,000, Abcam, Cambridge, UK). After washing, membranes were incubated with horseradish peroxidase-conjugated secondary antibodies at room temperature for 1 h. Protein bands were visualized using an enhanced chemiluminescence (ECL) detection reagent (Thermo Fisher Scientific) and quantified using ImageJ software (version 1.53g, NIH, Bethesda, MD, USA).

### 2.4. Animal Protocols

Six-week-old male ApoE^−/−^ mice (C57BL/6 background, Jackson Laboratory, Bar Harbor, ME, USA) weighing 20–25 g were housed in a specific pathogen-free environment under a 12 h light/dark cycle with free access to food and water at the Laboratory Animal Center of Ajou University. Experimental procedures were approved by the Institutional Animal Care and Use Committee (IACUC) of Ajou University (No. 23-3443).

To generate the diabetic atherosclerosis model, mice were randomly divided into the following three groups at 8 weeks of age: control, diabetic vehicle-treated (DM-vehicle), and diabetic imeglimin-treated (DM-imeglimin) groups. Mice in the DM-vehicle and DM-imeglimin groups were intraperitoneally injected with STZ (Sigma–Aldrich, Saint Louis, MO, USA) (55 mg·kg^−1^·d^−1^) for five consecutive days to induce diabetes, as previously reported [20]. STZ was dissolved in citrate buffer (pH 4.5). Two weeks after STZ administration, blood glucose levels were measured to assess diabetes induction. Blood glucose levels > 300 mg/dL were considered indicative of diabetes. No mice were excluded, as they all met this criterion. The control group was administered vehicle (citrate buffer) instead of STZ.

Imeglimin (Adooq Bioscience) was prepared in 0.1% methylcellulose and administered orally (gavage) at a dose of 50 mg/kg/day for 9 weeks. This dosage was selected based on prior research [21]. The control and DM-vehicle groups received the same volume of 0.1% methylcellulose. Body weight was monitored weekly (10–19 weeks of age), and at the study endpoint, mice were euthanized for sample collection

### 2.5. Blood Sample Assay

Blood was collected from the right ventricle of anesthetized mice after an 8 h fasting period. The collected blood was transferred into sterile 1.5 mL tubes and left at room temperature for 30 min. Samples were then centrifuged at 3000 rpm for 10 min to separate the serum, which was stored at −80 °C until analysis. The levels of serum glucose, total cholesterol, triglycerides, LDL and HDL cholesterols, and creatinine were measured using enzymatic assays on a Cobas c502 analyzer (Roche Diagnostics, Mannheim, Germany).

### 2.6. En Face Plaque Area Analysis

To evaluate atherosclerotic lesion formation, the entire aorta was excised post euthanasia. Briefly, after carefully removing both the perivascular connective and adipose tissues around the aorta, the aortic arch and thoracic to abdominal aorta were removed and opened longitudinally. Aortic tissues were stained using Oil Red O (Sigma–Aldrich) and imaged using a stereomicroscope. The percentage of plaque area was analyzed using ImageJ software (NIH Image, Bethesda, MD, USA).

### 2.7. Statistical Analysis

Data are presented as the means ± SEMs from a minimum of three independent experiments. Statistical analyses were performed using GraphPad Prism 7 (GraphPad Software Inc., San Diego, CA, USA). Differences between groups were analyzed using Student’s *t*-test. Statistical significance was set at *p* < 0.05.

## 3. Results

### 3.1. Imeglimin Inhibits Foam Cell Formation and Promotes Cholesterol Efflux in ox-LDL/HG-Treated THP-1 Macrophages

To assess the effect of imeglimin on foam cell formation and lipid accumulation in THP-1 macrophages, the cells were pretreated with imeglimin and then exposed to 100 μg/mL ox-LDL and 30 mM glucose. As shown in Figure 1, compared to the control, THP-1 macrophages stimulated with ox-LDL/HG showed higher lipid accumulation, as detected by Oil Red O staining. Notably, imeglimin treatment ameliorated intracellular lipid accumulation in THP-1 macrophages compared to that in ox-LDL/HG-treated cells. Subsequently, we investigated the effects of imeglimin on HDL-mediated cholesterol efflux in foam cells. The results showed that imeglimin significantly enhanced the cholesterol efflux (Figure 2).

### 3.2. Imeglimin Upregulates the Expression of ABCA1 and ABCG1 and Inhibits That of CD36 in ox-LDL/HG-Treated THP-1 Macrophages

Cholesterol efflux in macrophages is known to be regulated by ABCA1 and ABCG1. Therefore, we examined the effects of imeglimin on ABCA1 and ABCG1 expression levels. Imeglimin increased the expression of ABCA1 and ABCG1 proteins compared to that in ox-LDL/HG-treated THP-1 macrophages (Figure 3).

To determine whether imeglimin decreases ox-LDL uptake through downregulation of the expression of a scavenger receptor, we investigated the effect of imeglimin on CD36. As shown in Figure 3, imeglimin treatment normalized the ox-LDL/HG-induced increase in CD36 protein expression.

### 3.3. Imeglimin Regulates the Expression of ABCG1 and CD36 via Adenosine Monophosphate-Activated Protein Kinase (AMPK) Activation in ox-LDL/HG-Treated THP-1 Macrophages

Previous studies demonstrated that AMPK exerts antiatherosclerotic effects by upregulating ABCA1 and ABCG1 expression and promoting cholesterol efflux from macrophages [22,23]. Based on this evidence, we investigated whether imeglimin regulated cholesterol uptake and efflux through AMPK activation. We observed that imeglimin treatment enhanced AMPK phosphorylation in ox-LDL/HG-treated THP-1 macrophages (Figure 4A,B). In addition, imeglimin-induced upregulation of ABCG1 expression was abolished by pretreatment with compound C, an AMPK inhibitor, whereas ABCA1 expression did not show any statistically significant changes in the presence of compound C (Figure 4C,D). Furthermore, imeglimin-induced inhibition of CD36 protein expression was reversed by pretreatment with compound C (Figure 4C,D). These findings implied that imeglimin alleviated foam cell formation by increasing the levels of cholesterol efflux transporters (ABCG1) and suppressing CD36 expression through AMPK activation.

### 3.4. Effect of Imeglimin on Body Weight and Serum Biochemical Parameters in Diabetic ApoE^−/−^ Mice

The body weights of ApoE^−/−^ mice were measured weekly between 10 and 19 weeks of age, and no significant differences between mice from the Diabetes Mellitus (DM)-vehicle and DM-imeglimin groups were observed (Figure 5).

In addition, after nine weeks of imeglimin treatment, we measured mice fasting serum levels of glucose, lipids, and creatinine. We found that imeglimin treatment significantly reduced fasting glucose, total cholesterol, and LDL cholesterol levels in mice from the DM-imeglimin group compared to those in mice from the DM-vehicle group (Table 1). However, creatinine, HDL cholesterol, and triglyceride levels were not significantly different between the groups (Table 1). During the experimental period, no mice died.

### 3.5. Imeglimin Alleviates Atherosclerotic Plaque Formation in Diabetic ApoE^−/−^ Mice

En face analyses of Oil Red O-stained atherosclerotic plaques in the aortic arch and thoracic aorta of mice from the different groups demonstrated that the percentage area of atherosclerotic plaques was significantly increased in mice from the DM-vehicle group compared to that in mice from the control group. Moreover, imeglimin administration significantly reduced the atherosclerotic plaque area in mice from the DM-imeglimin group compared to that in mice from the DM-vehicle group (Figure 6).

### 3.6. Effects of Imeglimin on ABCA1, ABCG1, and CD36 Expression, and AMPK Activation in the Liver and Aorta of Diabetic ApoE^−/−^ Mice

To evaluate the impact of imeglimin on cholesterol uptake and efflux in the liver and aorta of diabetic ApoE^-/-^ mice, we analyzed AMPK activity and protein expression levels of ABCA1, ABCG1, and CD36 in these tissues. In the liver, imeglimin treatment significantly increased the expression levels of Thr-172 phosphorylation of AMPK, ABCA1, and ABCG1, while it significantly reduced CD36 protein levels in mice from the DM-imeglimin group compared to those in mice from the DM-vehicle group (Figure 7A,B). Imeglimin treatment also significantly upregulated ABCG1 protein expression in the aorta (Figure 7C,D).

## 4. Discussion

In this study, we showed for the first time that imeglimin inhibits macrophage foam cell formation by promoting cholesterol efflux via increasing the expression levels of ABCA1 and ABCG1 and decreasing cholesterol uptake by downregulating CD36 expression in ox-LDL/HG-treated THP-1 macrophages. The effects of imeglimin on the expression of ABCG1 and CD36 was mediated by AMPK. Furthermore, imeglimin showed a protective effect, limiting the development of atherosclerotic lesions in diabetic ApoE^−/−^ mice.

The study design of the in vitro and in vivo experiments was based on the respective pathophysiological contexts. To investigate the preventive effect of imeglimin, we assessed foam cell formation in vitro using THP-1 macrophages, while its therapeutic impact was evaluated in vivo using diabetic ApoE^−/−^ mice.

Foam cell formation is an essential pathological process in the early stages of atherosclerosis and plays an important role in its progression [4]. The main cause of foam cell formation is excessive influx of modified LDL, such as ox-LDL, and decreased cholesterol efflux. Therefore, we first determined the effects of imeglimin on THP-1 cell lipid uptake and cholesterol efflux. We found that imeglimin downregulates the expression of the scavenger receptor CD36 in ox-LDL/HG-treated THP-1 macrophages. Among the many members of the scavenger receptor family involved in foam cell formation, CD36 (a member of the class B family of scavenger receptors) is the primary mediator of ox-LDL uptake [24]. Therefore, CD36 plays an important role in foam cell formation [25]. Several studies have demonstrated that the deletion of CD36 in hyperlipidemic mice significantly reduces atherosclerosis and arterial lipid accumulation [26,27]. We also found that imeglimin increased ABCA1 and ABCG1 protein expression, indicating that imeglimin inhibited foam cell formation by increasing cholesterol efflux and decreasing cholesterol uptake in ox-LDL/HG-treated THP-1 macrophages.

Recent studies have demonstrated that imeglimin, owing to its structural similarity to metformin, activates AMPK in hepatocytes and 3T3-L1 adipocytes [28,29]. However, its ability to activate AMPK in macrophages remains unclear. AMPK is a key regulator of energy balance and metabolism [30]. In addition to its fundamental role in cellular energy regulation, AMPK is critically involved in metabolic disorders and inflammation-related diseases, including atherosclerosis [30,31]. Emerging evidence suggests that AMPK activation promotes cholesterol efflux from macrophages, thereby reducing foam cell formation [22,23]. In this study, we found that imeglimin treatment enhanced ABCG1-mediated cholesterol efflux and suppressed CD36-mediated cholesterol uptake via an AMPK-dependent signaling pathway in ox-LDL/HG-treated THP-1 macrophages. However, treatment with the AMPK inhibitor compound C did not significantly attenuate imeglimin-induced upregulation of ABCA1 expression, indicating that the regulation of ABCA1 by imeglimin is independent of AMPK. These findings align with those in previous reports demonstrating that treatment with AICAR, a pharmacological AMPK activator, increases ABCG1 protein expression in J774.A1 macrophages but does not affect ABCA1 expression [22].

In vivo experiments using diabetic ApoE^−/−^ mice demonstrated that imeglimin treatment reduced the atherosclerotic plaque area. Additionally, in the liver tissue, imeglimin activated AMPK and increased the expression of ABCG1 and ABCG1 proteins, while reducing CD36 expression. However, in the aorta, the effects of imeglimin on AMPK activation and ABCA1 and CD36 expression were limited, with a significant increase observed only in ABCG1 expression. AMPK activation in the liver has been reported to play an important role in maintaining hepatic lipid homeostasis [30]. Therefore, the pronounced activation of AMPK in the liver following imeglimin treatment may be attributed to its central role in the regulation of lipid metabolism. In contrast, this effect may be less evident in other tissues such as aortic tissues. Therefore, in our study, the differential effects of imeglimin on AMPK activation and ABCA1 and CD36 expression observed in the liver and aorta may be explained by tissue-specific variations.

We observed that imeglimin exhibited glucose-lowering and LDL cholesterol-lowering effects in diabetic ApoE^−/−^ mice, which likely contributed to the reduction in atherosclerosis observed in our study. Consistent with our findings, Sanada et al. recently reported a protective effect of imeglimin on atherosclerosis in streptozotocin (STZ)-induced diabetic ApoE^−/−^ mice [32]. However, they did not observe any significant effects of imeglimin on blood glucose or LDL cholesterol levels, leading them to propose that the antiatherosclerotic effects of this drug are independent of its glycemic and lipid control effects. The reason for these discrepancies remains unclear. While both studies employed STZ to induce diabetes in ApoE^−/−^ mice, the mean fasting blood glucose and LDL cholesterol levels measured in our study were relatively higher than those observed in the Sanada et al. animal model. These differences in metabolic profiles may have influenced the observed effects of imeglimin on blood glucose and LDL-cholesterol levels, potentially accounting for these divergent findings. However, previous randomized controlled trials in humans have reported that imeglimin does not significantly affect LDL cholesterol levels or cause a slight increase [9,11,33]. Given this, the LDL cholesterol reduction observed with imeglimin treatment in our study is likely specific to our experimental model and conditions.

In conclusion, this study provides new mechanistic insight into the antiatherogenic effects of imeglimin through the regulation of cholesterol uptake and efflux, at least in part, in an AMPK-dependent manner. These findings suggest that imeglimin may play a preventive role in foam cell formation and a therapeutic role in atherosclerosis progression under diabetic conditions.

## Figures and Tables

**Figure 1 cells-14-00472-f001:**
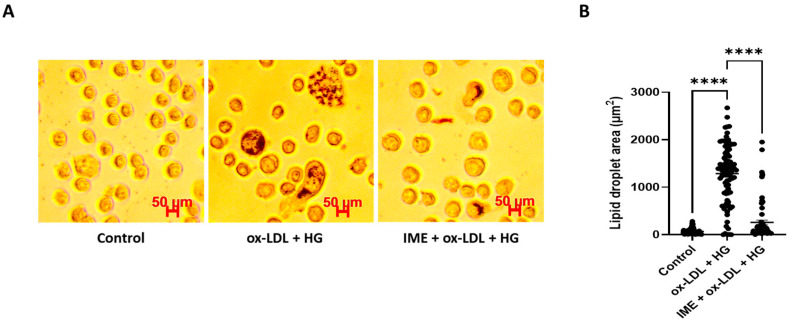
Effect of imeglimin on foam cell formation. THP-1 macrophages were pretreated with imeglimin and then exposed to 100 μg/mL ox-LDL and 30 mM glucose. (**A**) The formation of foam cells was detected by Oil Red O staining. (**B**) The lipid droplet area was quantified using Image J. Results are expressed as the means ± SEMs of at least three independent experiments. **** *p* < 0.0001. HG, high glucose; IME, imeglimin; ox-LDL, oxidized low-density lipoproteins.

**Figure 2 cells-14-00472-f002:**
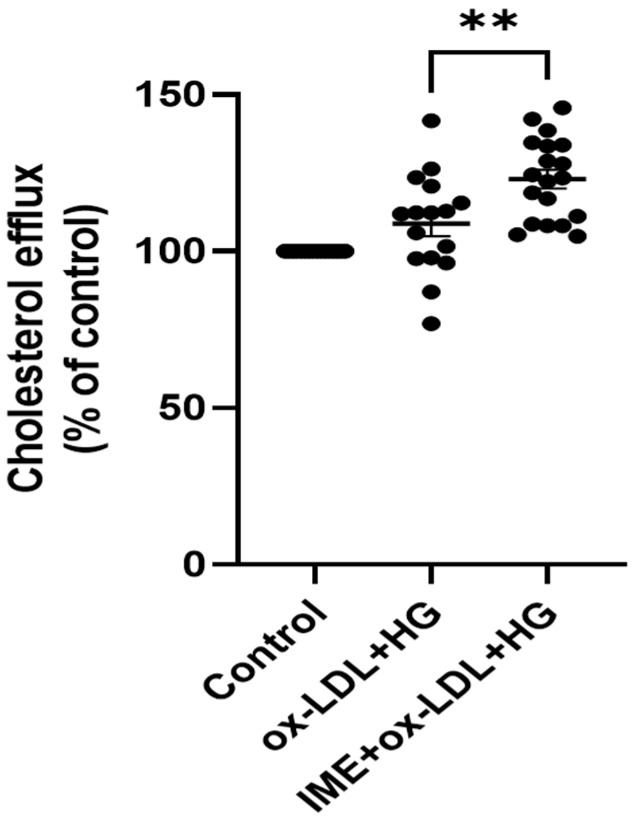
Effect of imeglimin on cholesterol efflux. THP-1 macrophages were pretreated with imeglimin and then exposed to 100 μg/mL ox-LDL and 30 mM glucose. Treatment with imeglimin promotes cholesterol efflux in THP-1 macrophage-derived foam cells. Cholesterol efflux was assayed using a fluorometric assay kit. Results are expressed as the means ± SEMs from five independent experiments, each performed in triplicate. ** *p* < 0.01.

**Figure 3 cells-14-00472-f003:**
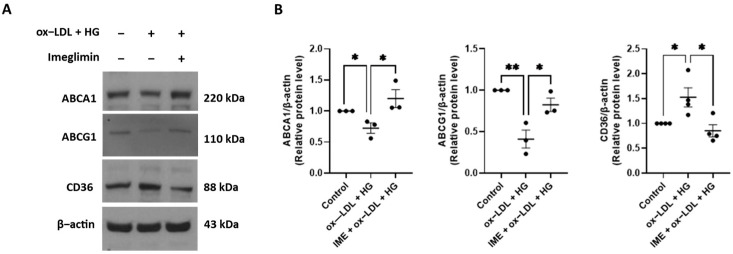
Effects of imeglimin on the expression of ABCA1, ABCG1, and CD36 proteins in ox-LDL/HG-treated THP1 macrophages. THP-1 macrophages were pretreated with imeglimin and then exposed to 100 μg/mL ox-LDL and 30 mM glucose. (**A**) Western blot analysis showing the expression of ABCA1, ABCG1, and CD36 proteins; (**B**) Relative protein expression levels of ABCA1, ABCG1, and CD36 (quantified using ImageJ). Results are expressed as the means ± SEMs of at least three independent experiments. * *p* < 0.05, ** *p* < 0.01. HG, high glucose; IME, imeglimin; oxLDL, oxidized lowdensity lipoproteins.

**Figure 4 cells-14-00472-f004:**
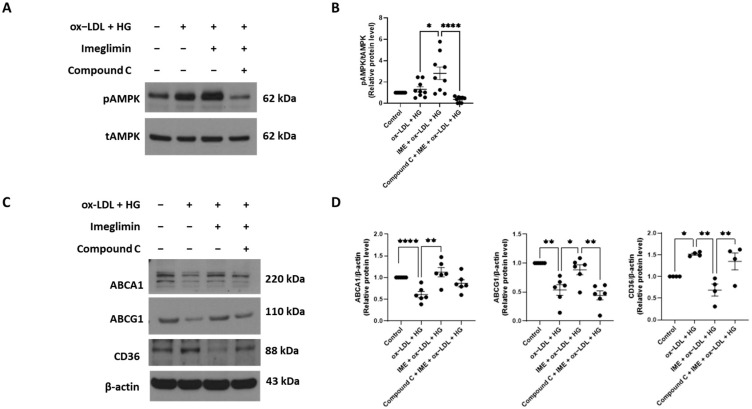
Effect of imeglimin on the expression of ABCA1, ABCG1, and CD36 in the presence of the AMPK inhibitor compound C. THP-1-derived macrophages were pretreated with compound C, incubated with 100 μM imeglimin for 24 h, and exposed to 100 μg/mL oxLDL and 30 mM glucose. (**A**) Western blot analysis showing the expression of total AMPK (tAMPK) and phosphorylated AMPK (*p*-AMPK) proteins; (**B**) Relative protein expression levels of t-AMPK and pAMPK (quantified using ImageJ); (**C**) Western blot analysis showing the expression of ABCA1, ABCG1, and CD36 proteins; (**D**) Relative protein expression levels of ABCA1, ABCG1, and CD36 (quantified using ImageJ). Results are expressed as the means ± SEMs of at least three independent experiments. * *p* < 0.05, ** *p* < 0.01, **** *p* < 0.0001. HG, high glucose; IME, imeglimin; ox-LDL, oxidized low-density lipoproteins.

**Figure 5 cells-14-00472-f005:**
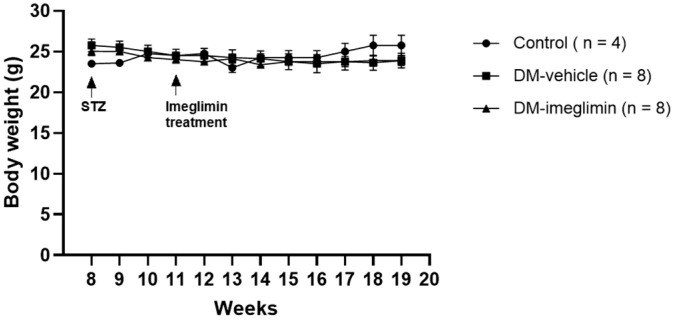
Body weight changes in mice after nine weeks of imeglimin treatment. Diabetic mice show a decrease in body weight over time compared to control ApoE^−/−^ mice. No significant difference in body weight is observed between diabetic mice from the DMvehicle and DM-imeglimin groups. STZ, streptozotocin.

**Figure 6 cells-14-00472-f006:**
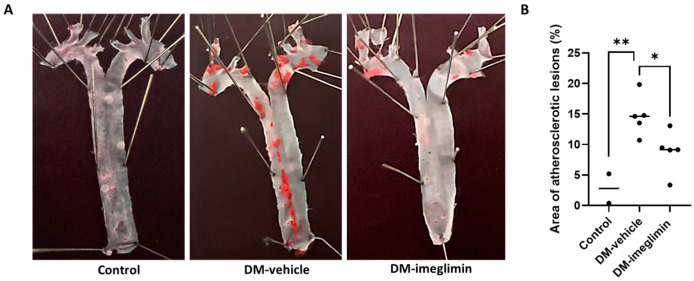
Effect of imeglimin on atherosclerotic plaque formation in the aortas of diabetic ApoE^−/−^ mice. (**A**) Representative images of en face analyses of mice aortas showing atherosclerotic plaques stained with Oil Red O; (**B**) Quantitative measurement of the en face plaque area (%) observed in mice aortas stained with Oil Red O. Results are expressed as the means ± SEMs of independent experiments (Control: *n* = 2, DM-vehicle: *n* = 5, DM-imeglimin: *n* = 5). * *p* < 0.05, ** *p* < 0.01.

**Figure 7 cells-14-00472-f007:**
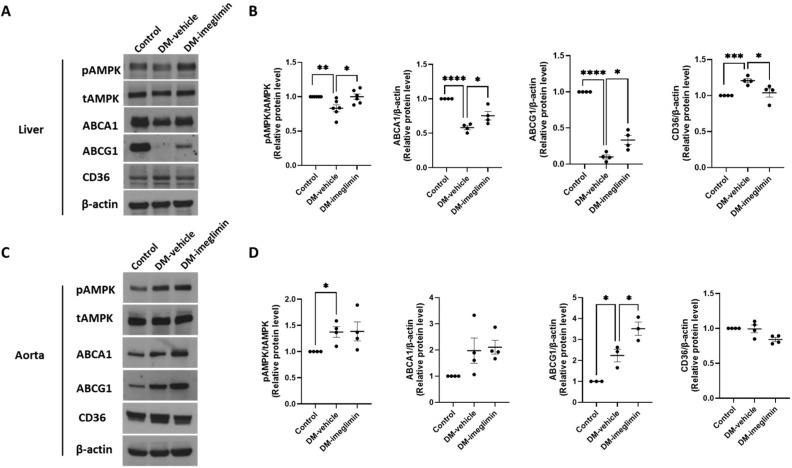
Effects of imeglimin on AMPK activity and ABCA1, ABCG1, and CD36 protein expression in diabetic ApoE^-/-^ mice. (**A**) Western blot analysis showing the expression of pAMPK, tAMPK, ABCA1, ABCG1, and CD36 in the liver; (**B**) Relative protein expression levels of pAMPK, tAMPK, ABCA1, ABCG1, and CD36 in the liver (quantified using ImageJ); (**C**) Western blot analysis showing the expression of pAMPK, tAMPK, ABCA1, ABCG1, and CD36 in the aorta; (**D**) Relative protein expression levels of pAMPK, tAMPK, ABCA1, ABCG1, and CD36 in the aorta (quantified using ImageJ). Results are expressed as the means ± SEMs of at least three independent experiments. * *p* < 0.05, ** *p* < 0.01, *** *p* < 0.001, **** *p* < 0.0001.

**Table 1 cells-14-00472-t001:** Serum biochemical profiles of mice from this study.

	Control Group (*n* = 4)	DM-Vehicle Group (*n* = 8)	DM-Imeglimin Group (*n* = 8)
Fasting glucose level (mg/dL)	300 ± 27.58	712.4 ± 27.35	606.6 ± 27.74 *
Total cholesterol (mg/dL)	540 ± 39.53	1257 ± 61.20	935.5 ± 10.8 *
Triglyceride (mg/dL)	109.5 ± 11.85	134.5 ± 15.73	98.63 ± 23.80
HDL cholesterol (mg/dL)	48.25 ± 3.065	110.5 ± 11.01	84.13 ± 10.64
LDL cholesterol (mg/dL)	245.5 ± 17.40	962.3 ± 60.93	697.3 ± 83.07 *
Creatinine (mg/dL)	0.175 ± 0.0104	0.2588 ± 0.0166	0.2263 ± 0.0169

Data are shown as mean ± SEM. * *p* < 0.05 compared to the DM-vehicle group.

## Data Availability

The data supporting the findings of this study are all included in the manuscript.

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
