# Peer review of "Imeglimin Inhibits Macrophage Foam Cell Formation and Atherosclerosis in Streptozotocin-Induced Diabetic ApoE-Deficient Mice"

_cells, 2025, doi:10.3390/cells14070472_

Round 1
Reviewer 1 Report
Comments and Suggestions for Authors
The authors aimed to elucidate the therapeutic potential of imeglimin in diabetic cardiovascular disease by investigating in vitro and in vivo the effects of this drug on foam cell formation and atherosclerosis.
The study is interesting and could give hints on the hypothetical antiatherosclerotic effect of imeglimin. The manuscript is well-written and the experiments are convincing. However, I still have some comments to make.
Lines 44-45: It should read cholesterol efflux from the macrophages.
Lines 176-177: The experiments shown in Figure 1 do not demonstrate that Imglimin inhibits intracellular lipid deposition by increasing HDL-mediated cholesterol efflux from THP-1 macrophages since the experimental conditions for figure 1A are different from what has been done for Figure 1B. To compare the two situation, the authors should incubate the cells with HDL also when they try to upload cholesterol by using ox-LDL. The data shown in Figure 1A only proofs that imeglimin most probably blocks ox-LDL uptake by THP-1 cells. The authors must check this. Then they may try to formulate a hypothesis regarding the mechanism.
Lines 194-195: This is the right experiment I was talking about before. I do not understand why the authors stated wrongly in the previous sentence. However, I would not say that imeglimin decreased CD36 expression, but that it normalized imeglimin induced expression by ox-LDL+HG.
The observation that imeglimin reduces LDL-cholesterol levels is interesting but I would like also to see what happens to the LDL receptor after imeglimin treatment of the apoE mice. Is it upregulated? This data would give a more clear perspective of imeglimin mechanism of action.
Author Response
Comments 1:
Lines 44-45: It should read cholesterol efflux from the macrophages.
Response 1:
We appreciate the reviewer’s careful reading of our manuscript. We have corrected the sentence to ‘cholesterol efflux from the macrophages’ as suggested.
Comments 2:
Lines 176-177: The experiments shown in Figure 1 do not demonstrate that Imglimin inhibits intracellular lipid deposition by increasing HDL-mediated cholesterol efflux from THP-1 macrophages since the experimental conditions for figure 1A are different from what has been done for Figure 1B. To compare the two situation, the authors should incubate the cells with HDL also when they try to upload cholesterol by using ox-LDL. The data shown in Figure 1A only proofs that imeglimin most probably blocks ox-LDL uptake by THP-1 cells. The authors must check this. Then they may try to formulate a hypothesis regarding the mechanism.
Response 2:
We appreciate the reviewer’s insightful comment. We acknowledge that presenting Figure 1A and Figure 1B together may have led to some confusion in interpreting the results. As the reviewer pointed out, Figure 1A demonstrates that imeglimin reduces foam cell formation. However, this result alone does not directly indicate whether the reduction is due to decreased ox-LDL uptake or increased cholesterol efflux. In contrast, Figure 1B provides evidence that imeglimin enhances HDL-mediated cholesterol efflux.
To improve clarity and avoid misinterpretation, we have revised the figures by separating Figure 1A into Figure 1, with the addition of a statistical analysis graph (as per another reviewer's suggestion), and Figure 1B into Figure 2. Additionally, we agree with the reviewer that the statement in Lines 176–177 ("These results demonstrated that imeglimin inhibited intracellular lipid deposition by increasing HDL-mediated cholesterol efflux in THP-1 macrophages.") did not accurately describe the results of Figure 1. Therefore, we have removed this sentence.
Our investigation into the effect of cholesterol uptake of imeglimin was presented later, supported by CD36 expression data. We found that imeglimin normalized the expression of CD36, a key scavenger receptor involved in ox-LDL uptake, which was upregulated in response to ox-LDL/HG stimulation. Taken together, these findings suggest that imeglimin inhibits foam cell formation in ox-LDL/HG-treated THP-1 cells by reducing ox-LDL uptake through CD36 downregulation and enhancing HDL-mediated cholesterol efflux.
We hope these modifications resolve your concerns.
Comments 3:
Lines 194-195: This is the right experiment I was talking about before. I do not understand why the authors stated wrongly in the previous sentence. However, I would not say that imeglimin decreased CD36 expression, but that it normalized imeglimin induced expression by ox-LDL+HG.
Response 3:
We appreciate the reviewer’s insightful comments. We acknowledge the need for a more precise description and have revised the sentence accordingly. The sentence now reads: "Imeglimin treatment normalized the ox-LDL/HG-induced increase in CD36 protein expression."
Thank you for your valuable suggestion, which has helped improve the clarity and accuracy of our manuscript.
Comments 4:
The observation that imeglimin reduces LDL-cholesterol levels is interesting but I would like also to see what happens to the LDL receptor after imeglimin treatment of the apoE mice. Is it upregulated? This data would give a more clear perspective of imeglimin mechanism of action.
Response 4:
We sincerely appreciate the reviewer’s insightful comment regarding the potential effects of imeglimin on LDL cholesterol levels and LDL receptor regulation. Previous randomized controlled trials in humans have reported that imeglimin does not significantly affect LDL cholesterol levels or cause a slight increase (Reference 1,2,3) The observed LDL cholesterol-lowering effect of imeglimin in our study was specific to our experimental model using diabetic ApoE-knockout mice, and this may have contributed, at least in part, to the observed anti-atherosclerotic effects. Therefore, we believe that the reduction in LDL cholesterol observed in our study is likely a phenomenon specific to our experimental model and conditions, rather than a broadly applicable effect of imeglimin on lipid metabolism. To provide further clarification, we have addressed this point in the revised discussion as follows.
However, previous randomized controlled trials in humans have reported that imeglimin does not significantly affect LDL cholesterol levels or cause a slight increase [10,11,32]. Given this, the LDL cholesterol reduction observed with imeglimin treatment in our study is likely specific to our experimental model and conditions.
Furthermore, the primary objective of our study was to investigate the mechanism by which imeglimin inhibits foam cell formation and atherosclerosis. While the regulation of the LDL receptor is an interesting aspect, we believe that further investigation into this pathway falls beyond the intended scope of our study.
We hope this explanation clarifies our approach, and we sincerely appreciate the reviewer’s valuable feedback.
Reference
- Dubourg, J.; Fouqueray, P.; Thang, C.; Grouin, J.M.; Ueki, K. Efficacy and Safety of Imeglimin Monotherapy Versus Placebo in Japanese Patients With Type 2 Diabetes (TIMES 1): A Double-Blind, Randomized, Placebo-Controlled, Parallel-Group, Multicenter Phase 3 Trial. Diabetes Care 2021, 44, 952-959, doi:10.2337/dc20-0763.
- Reilhac, C.; Dubourg, J.; Thang, C.; Grouin, J.M.; Fouqueray, P.; Watada, H. Efficacy and safety of imeglimin add-on to insulin monotherapy in Japanese patients with type 2 diabetes (TIMES 3): A randomized, double-blind, placebo-controlled phase 3 trial with a 36-week open-label extension period. Diabetes Obes Metab 2022, 24, 838-848, doi:10.1111/dom.14642.
- Dubourg, J.; Ueki, K.; Grouin, J.M.; Fouqueray, P. Efficacy and safety of imeglimin in Japanese patients with type 2 diabetes: A 24-week, randomized, double-blind, placebo-controlled, dose-ranging phase 2b trial. Diabetes Obes Metab 2021, 23, 800-810, doi:10.1111/dom.14285.
Reviewer 2 Report
Comments and Suggestions for Authors
In the present MS, authors Ji Yeon Lee et collaborators investigated in vitro and in vivo impacts of imeglimin drug on cultured foam cell formation and on atherosclerotic plaques formation in a murine model.
Authors tested hypothesis that imeglimin, antidiabetic drug may exert antiatherogenic properties.
By using valuable techniques, Authors showed improved expression of actors implicated in cholesterol efflux and foam cell formation when macrophage (cultured in an atherogenic media) were treated with imeglimin. In vivo, a reduction of atherosclerotic plaque lesion was observed in diabetic Apo E deficient mice when treated with imeglimin.
This is an interesting MS dealing with a subject of interest.
That said I have point that should be considered by authors:
- Introduction
Paragraph starting with “Imeglimin is the first member of a new class of oral glucose-lowering agents…” I think authors should specify whether imeglimin is approved for diabetes treatment. The authors should mention results and cite clinical studies using the molecule.
- M and M section
Bottom of page 3. Authors should specify how many streptozotocin-treated mice were excluded in the study due to glucose levels inferior to 300 mg/dL.
I noticed that preventive impact of imeglimin was tested on foam cell formation whereas therapeutic impact of imeglimin was tested in diabetic mice. Can authors explain this discrepancy?
- Results
Figure 1A. Authors should provide a quantification of lipid accumulation in cells together with statistical analysis.
Figure 1A is missing a scale (small bar corresponding to 50 µm).
Figure 1B Authors should include number of samples in the figure legend together with separate dots on the graph instead of histograms.
Figure 2B Authors should include separate dots on the graph instead of histograms.
I noticed the presence of multiple extra bands for ABCA1, CD36 on western blots (original and uncropped images). What does it mean?
Figure 4, the number of mice is not specified.
Table 1. Why is the number of mice in the control group different?
Figure 5, the number of mice is not specified. Again, I think it would be better to see separate dots.
Same remark for figure 6.
Author Response
Comments 1:
- Introduction
Paragraph starting with “Imeglimin is the first member of a new class of oral glucose-lowering agents…” I think authors should specify whether imeglimin is approved for diabetes treatment. The authors should mention results and cite clinical studies using the molecule.
Response 1:
We agree that providing further clarification regarding the approval status of imeglimin and its clinical studies would be valuable. We have added this information as follows:
Imeglimin, a novel oral glucose-lowering agent, was approved for the treatment of type 2 diabetes in Japan in 2021[8]. It is the first drug in the glimins class and is marketed under the brand name TWYMEEG®. The approval was based on the results of several clinical trials, including pivotal Phase III studies, which demonstrated its ef-ficacy and safety in managing blood glucose levels both as a monotherapy and in com-bination with other antidiabetic drugs [9-11]
Comments 2:
- M and M section
Bottom of page 3. Authors should specify how many streptozotocin-treated mice were excluded in the study due to glucose levels inferior to 300 mg/dL.
Response 2:
No mice were excluded from the study based on glucose levels, as all streptozotocin-treated mice had blood glucose levels exceeding 300 mg/dL. We have included this information in the revised manuscript.
Comments 3:
I noticed that preventive impact of imeglimin was tested on foam cell formation whereas therapeutic impact of imeglimin was tested in diabetic mice. Can authors explain this discrepancy?
Response 3:
We appreciate the reviewer’s insightful comment. The study design of the in vitro and in vivo experiments was based on the respective pathophysiological contexts.
In our in vitro model, we aimed to mimic a diabetic-like condition by exposing cells to both ox-LDL and high glucose. We focused on the preventive impact of imeglimin by pre-treating macrophages before ox-LDL and high glucose exposure to assess its ability to mitigate foam cell formation. This approach was chosen to specifically examine the early protective mechanisms of imeglimin at the cellular level.
The in vivo study aimed to evaluate the therapeutic potential of imeglimin in an established diabetic atherosclerosis model, reflecting a more clinically relevant scenario where treatment is initiated after disease onset. This design takes into account the clinical use of imeglimin as an oral anti-diabetic agent and explores its potential to reduce atheroma in a diabetic environment.
Thus, our study seeks to provide insights into the protective and therapeutic effects of imeglimin in the context of diabetic atherosclerosis. We hope this explanation clarifies our study design, and we sincerely appreciate the reviewer’s valuable input.
Comments 4:
- Results
Figure 1A. Authors should provide a quantification of lipid accumulation in cells together with statistical analysis.
Response 4:
We have added this information in Figure 1.
Comments 5:
Figure 1A is missing a scale (small bar corresponding to 50 µm).
Response 5:
The scale bars were already present in the figure, but it seems to have been difficult to see. We have now highlighted it by making it thicker for better visibility.
Comments 6:
Figure 1B Authors should include number of samples in the figure legend together with separate dots on the graph instead of histograms.
Response 6:
We have modified Figure 1B to display separate dot plots, and the number of samples has been included in the figure legend. Additionally, as per another reviewer's suggestion, we have separated Figure 1B into a new Figure 2.
Comments 7:
Figure 2B Authors should include separate dots on the graph instead of histograms.
Response 7:
According to the reviewer’s suggestion, we have modified Figure 2B to display separate dot plots. (Figure 1B has been separated into a new Figure 2, and the original Figure 2 has been renumbered as Figure 3)
Comments 8:
I noticed the presence of multiple extra bands for ABCA1, CD36 on western blots (original and uncropped images). What does it mean?
Response 8:
The presence of multiple bands in the Western blot for ABCA1 and CD36 is due to the characteristics of the antibody. For ABCA1, we used the Novus antibody, which the manufacturer specifies as follows: 'The Western blot band representing ABCA1 is observed at approximately 220 kDa. Additional non-specific bands appear at lower molecular weights but do not interfere with the ABCA1 signal’. In addition, CD36 is a heavily glycosylated plasma membrane glycoprotein, which is the primary reason for the appearance of multiple bands of varying molecular weights. According to antibody testing data on the manufacturer's website (ThermoFisher Scientific) and published literature using this antibody, additional bands have been consistently observed. Thus, the additional bands observed in our study are consistent with previously reported patterns for these antibodies and do not affect the interpretation of ABCA1 and CD36 expression levels.
We have added references below that reported multiple band patterns of ABCA1 and CD36.
1) J Am Heart Assoc. 2017 Apr 28;6(5):e005520. doi: 10.1161/JAHA.117.005520
2) Elife.2015 Jul 14:4e08009. doi:10.7554/eLife.08009.
3) Lipids Health Dis. 2013 Dec 8:12:180. doi: 10.1186/1476-511X-12-180
4) Am J Physiol Renal Physiol. 2024 Feb 1;326(2):F265-F277. doi: 0.1152/ajprenal.00154.2023
5) J Pers Med. 2021 Apr 7;11(4):278. doi: 10.3390/jpm11040278.
6) FASEB J. 2023 Apr;37(4):e22846. doi: 10.1096/fj.202201469R.
Comments 9:
Figure 4, the number of mice is not specified.
Response 9:
We have indicated the number of mice in figure 5.(the original Figure 4 has been renumbered as Figure 5)
Comments 10:
Table 1. Why is the number of mice in the control group different?
Response 10:
In our study, the primary objective was to compare the DM + imeglimin treatment group with the DM + vehicle group. Based on this goal, we intentionally included a smaller control group.
Comments 11:
Figure 5, the number of mice is not specified. Again, I think it would be better to see separate dots.
Same remark for figure 6.
Response 11:
According to the reviewer’s suggestion, the graphs originally presented as histograms have been modified to separate dot plots.
Round 2
Reviewer 1 Report
Comments and Suggestions for Authors
The authors replied to my comments.
Author Response
Comments 1 :
The authors replied to my comments.
Response 1:
We appreciate your time and feedback.
Reviewer 2 Report
Comments and Suggestions for Authors
In their revised version of their MS, Authors addressed most of my points.
Still, I am not convinced by their answer to my following point:
“I noticed that preventive impact of imeglimin was tested on foam cell formation whereas therapeutic impact of imeglimin was tested in diabetic mice. Can authors explain this discrepancy?”
If authors agreed in their response that preventive impact of imeglimin was tested in vitro on foam cell formation whereas therapeutic impact of imeglimin was tested in vivo, this fact was never specified in the MS.
More importantly, in the abstract the word therapeutical impact/result was written 2 times and authors conclude their MS with “These findings highlight the potential therapeutic
value of imeglimin”.
Author Response
Comments 1:
In their revised version of their MS, Authors addressed most of my points.
Still, I am not convinced by their answer to my following point:
“I noticed that preventive impact of imeglimin was tested on foam cell formation whereas therapeutic impact of imeglimin was tested in diabetic mice. Can authors explain this discrepancy?”
If authors agreed in their response that preventive impact of imeglimin was tested in vitro on foam cell formation whereas therapeutic impact of imeglimin was tested in vivo, this fact was never specified in the MS.
More importantly, in the abstract the word therapeutical impact/result was written 2 times and authors conclude their MS with “These findings highlight the potential therapeutic
value of imeglimin”.
Response 1:
We sincerely appreciate the reviewer’s continued efforts to improve our manuscript. We acknowledge the reviewer’s concern that our manuscript did not clearly distinguish between the preventive effect of imeglimin, which was evaluated in in vitro foam cell formation experiments, and its therapeutic effect, which was assessed in in vivo diabetic mice.
To clarify this point, we have now explicitly stated in Discussion section as follows:
The study design of the in vitro and in vivo experiments was based on the respective pathophysiological contexts. To investigate the preventive effect of imeglimin, we assessed foam cell formation in vitro using THP-1 macrophages, while its therapeutic impact was evaluated in vivo using diabetic ApoE -/- mice.
Additionally, as the reviewer pointed out, the term "therapeutic" used in the Abstract and Conclusion sections has been appropriately revised to prevent potential misinterpretation.
Abstract section
Page 1, line 14~16. This study aimed to elucidate the therapeutic potential of imeglimin in diabetic cardiovascular disease by investigating the effects of this drug on foam cell formation and atherosclerosis.
The revised sentence now reads: This study aimed to investigate the effects of imeglimin on foam cell formation and atherosclerosis in the context of diabetes.
Page 1, line 26~27. These results highlight the therapeutic potential of imeglimin in antiatherosclerosis treatments
The revised sentence now reads: These findings suggest that imeglimin may have a preventive effect on foam cell formation and a therapeutic role in atherosclerosis progression in diabetic conditions.
Conclusion section
These findings highlight the potential therapeutic value of imeglimin for mitigating the progression of atherosclerosis and associated cardiovascular diseases.
The revised sentence now reads: These findings suggest that imeglimin may play a preventive role in foam cell formation and a therapeutic role in atherosclerosis progression under diabetic conditions.
We hope these revisions adequately address your concern. Thank you once again for your insightful feedback, which has helped improve the clarity and precision of our manuscript.
Round 3
Reviewer 2 Report
Comments and Suggestions for Authors
Authors correctly addressed points I raised.